# Revisiting the guidelines for ending isolation for COVID-19 patients

**Yong Dam Jeong**[1,2†]**, Keisuke Ejima**[3†]*****, Kwang Su Kim**[1†]**, Shoya Iwanami**[1]**, Ana I Bento**[3]**, Yasuhisa Fujita**[1]**, Il Hyo Jung**[2]**, Kazuyuki Aihara**[4]**, Koichi Watashi**[5,6,7]**, Taiga Miyazaki**[8,9]**, Takaji Wakita**[5]**, Shingo Iwami**[1,10,11,12,13]*****, Marco Ajelli**[3,14]

[1]interdisciplinary Biology Laboratory (iBLab), Division of Biological Science, Graduate School of Science, Nagoya University, Nagoya, Japan; [2]Department of Mathematics, Pusan National University, Busan, Republic of Korea; [3]Department of Epidemiology and Biostatistics, Indiana University School of Public Health-Bloomington, Bloomington, United States; [4]International Research Center for Neurointelligence, The University of Tokyo, Tokyo, Japan; [5]Department of Virology II, National Institute of Infectious Diseases, Tokyo, Japan; [6]Research Center for Drug and Vaccine Development, National Institute of Infectious Diseases, Tokyo, Japan; [7]Department of Applied Biological Science, Tokyo University of Science, Noda, Japan; [8]Department of Infectious Diseases, Nagasaki University Graduate School of Biomedical Sciences, Nagasaki, Japan; [9]Division of Respirology, Rheumatology, Infectious Diseases, and Neurology, Department of Internal Medicine, Faculty of Medicine, University of Miyazaki, Miyazaki, Japan; [10]Institute of Mathematics for Industry, Kyushu University, Fukuoka, Japan; [11]Institute for the Advanced Study of Human Biology (ASHBi), Kyoto University, Kyoto, Japan; [12]NEXT-Ganken Program, Japanese Foundation for Cancer Research (JFCR), Tokyo, Japan; [13]Science Groove Inc, Fukuoka, Japan; [14]Laboratory for the Modeling of Biological and Socio-technical Systems, Northeastern University, Boston, United States

*For correspondence:
kejima@iu.edu (KE);
iwami.iblab@bio.nagoya-u.ac.jp
(SI)

[†]These authors contributed
equally to this work

**Competing interest:** See
page 12

**Reviewing editor:** Joshua T
Schiffer, Fred Hutchinson Cancer
Research Center, United States

**Abstract** Since the start of the COVID-19 pandemic, two mainstream guidelines for defining when to end the isolation of SARS-CoV-2-infected individuals have been in use: the one-size-fits-all approach (i.e. patients are isolated for a fixed number of days) and the personalized approach (i.e. based on repeated testing of isolated patients). We use a mathematical framework to model within-host viral dynamics and test different criteria for ending isolation. By considering a fixed time of 10 days since symptom onset as the criterion for ending isolation, we estimated that the risk of releasing an individual who is still infectious is low (0–6.6%). However, this policy entails lengthy unnecessary isolations (4.8–8.3 days). In contrast, by using a personalized strategy, similar low risks can be reached with shorter prolonged isolations. The obtained findings provide a scientific rationale for policies on ending the isolation of SARS-CoV-2-infected individuals.

## Introduction

Since the first case of a novel coronavirus (SARS-CoV-2) was identified in China in December of 2019, its associated disease, COVID-19, spread quickly around the world, with the number of cases reaching 80 million by the end of 2020. During this time, nonpharmaceutical interventions (NPIs) were used on a massive scale to suppress or mitigate SARS-CoV-2 transmission (*Cowling et al., 2020*). As of January 2021, several countries had started vaccination campaigns aimed at controlling SARS-CoV-2 spread (*Centers for Disease Control and Prevention, 2021*). Still, until such

vaccination programs reach a sizable fraction of the population, NPIs will likely continue to play a crucial role for epidemic control (*Yang et al., 2021a*).

A simple but effective NPI is the isolation of SARS-CoV-2-infected individuals. This can be done either in the infected person's place of residence (as is the case for most Western countries [*European Centre for Disease Prevention and Control, 2020b*]) or in dedicated facilities (as is the case in China [*Burki, 2020*]). In both cases, a criterion for determining when to end the isolation phase is needed. Although a longer isolation period may decrease the chance of transmission, it also entails both a higher burden on the mental and physical health of the patient (*Mian et al., 2021*) and cause higher economic loss (*Ash et al., 2021*). Scientifically sound guidelines for determining when to end isolation are thus warranted.

So far, two main approaches have been adopted by countries around the globe. The first approach is to isolate patients for a fixed time period (i.e. a one-size-fits-all approach). For example, the Centers for Disease Control and Prevention (CDC) created guidelines for health care practitioners concerning the discontinuation of transmission-based precautions for COVID-19 patients in health care settings that are based on the time since symptom onset or disappearance (i.e. symptom-based strategy) (*Centers for Disease Control and Prevention, 2020a*). In the CDC guidelines, those with mild to moderate illness can end isolation (or precautions) when the following three conditions are met: 'At least 10 days have passed since symptoms first appeared,' 'At least 24 hr have passed since last fever without the use of fever-reducing medications,' and 'Symptoms (e.g. cough, shortness of breath) have improved.' However, such a one-size-fits-all approach does not account for the individual variability in viral load (*Iwanami et al., 2020*), which is associated with both severity (*Zheng et al., 2020*) and persistence of symptoms (*Long et al., 2020*), and may thus not fully prevent further transmission.

The second approach is based on the assessment of the viral load of each isolated patient (i.e. personalized approach), and isolation ends when the viral load drops below a certain threshold value, which is associated with a low risk of further spreading the pathogen (*He et al., 2020*). The viral load can be measured by reverse transcription polymerase chain reaction (PCR), which can be used not only for diagnosing infection but also in determining when to end the isolation period. As an example, the CDC recommends using PCR testing in particular circumstances, such as for patients with severe immunodeficiency. The guidelines include both the resolution of symptoms and PCR test results, that is, 'Results are negative from at least two consecutive respiratory specimens collected ≥24 hr apart' (*Centers for Disease Control and Prevention, 2020a*).

The purpose of this study was to assess whether the personalized approach based on PCR test results minimizes the length of the isolation period while limiting the risk of prematurely releasing infectious individuals as compared with the one-size-fits-all approach. Moreover, we define best practices for the use of a PCR-based personalized approach. To do so, we developed a mathematical model of SARS-CoV-2 viral load dynamics (*Ejima et al., 2020*; *Iwanami et al., 2020*) that accounts for individual heterogeneity and is calibrated on longitudinal viral load data.

## Results

### Descriptive statistics

We identified four papers meeting the inclusion criteria (*Kim et al., 2020a*; *Wölfel et al., 2020*; *Young et al., 2020*; *Zou et al., 2020*). Among the patients reported in these four studies, 30 patients (approximately 60% of the participants in the original studies) met our inclusion criteria (*Table 1*). Three studies were from Asia and one was from Europe. The lowest and highest detection limit among those studies were 15.3 copies/mL and 68 copies/mL, respectively. These are relatively lower than the commonly used threshold values (the median was 100 copies/mL [*Fung et al., 2020*; *Giri et al., 2021*; *van Kasteren et al., 2020*]). The data were collected by February of 2020, which was during the early phase of the COVID-19 pandemic. Participants were hospitalized patients of ages ranging from 28 to 78 years; the sex ratio was mostly even.

### Model fitting

Three models were fitted to the data: the baseline model, the 'eclipse phase' model, and the 'innate immune response' model. The estimated model parameters, the estimated (mean) curves and the

**Table 1.** Summary of the viral load data used for modeling.

| Source | Country | # of included (excluded) patients | Sampling site | Reporting unit | Detection limit (copies/mL) | Symptom onset of patients | Age[‡] | Sex (M:F) |
|---|---|---|---|---|---|---|---|---|
| Young et al. | Singapore | 12 (6) | nasopharynx | cycle threshold[*] | 68.0 | 1/21 - 1/30 | 37.5 (31–56) | 6:6 |
| Zou et al. | China | 8 (8) | nose | cycle threshold[*] | 15.3 | 1/11 - 1/26 | 52.5 (28–78) | 3:5 |
| Kim et al. | Korea | 2 (7) | nasopharynx and oropharynx | cycle threshold[*] | 68.0 | NA | NA | NA |
| Wölfel et al. | Germany | 8 (1) | pharynx | viral load (copies/ swab)[†] | 33.3 | 1/23 - 2/4 | NA | NA |

*Viral load was calculated from cycle threshold values using the conversion formula: $\log_{10}(\text{Viral load [copies/mL]}) = -0.32 \times \text{Ct values [cycles]} + 14.11$ (**Peiris et al., 2003**).

† One swab = 3 mL (**Wölfel et al., 2020**).

‡ Median (range).

individual fitted curves are reported in *Supplementary file 1*, *Figure 1*, respectively. Although all the three models lead to similar results (*Figure 1*), the baseline model shows a longer tail than the two other models, due to the lower estimated death rate of infected cells (*Supplementary file 1*).

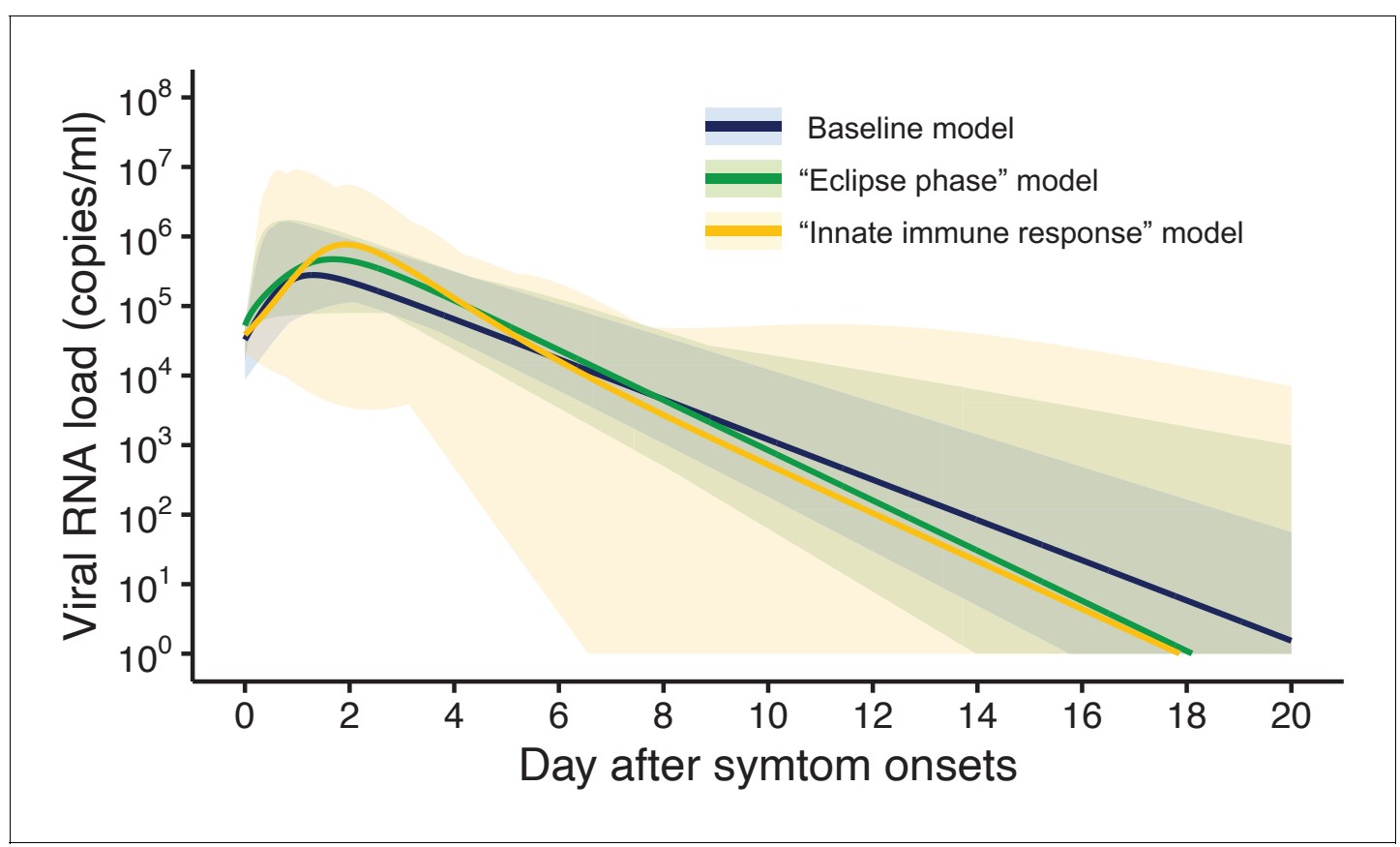

**Figure 1.** Estimated viral load curves from the three analyzed models. The solid lines are the estimated viral load curves of the three models for the best fit parameters (Blue: baseline model, Green: 'ecliplse phase' model, Yellow: 'innate immune response' model). The shaded regions correspond to 95% predictive intervals. The 95% predictive interval was created using bootstrap approach.

The online version of this article includes the following source data and figure supplement(s) for figure 1:

**Source data 1.** Estimated viral load curves from the three analyzed models.

**Figure supplement 1.** Observed and estimated viral load for individual patients.

Further, the three models showed similar values of the Akaike information criterion (AIC) and the Bayesian information criterion (BIC) (*Supplementary file 2*). Unless otherwise stated, the results presented in thereafter refer to the baseline model.

## One-size-fits-all approach

By considering a fixed time of 10 days since symptom onset as the criterion for ending isolation, the probability of releasing patients who are still infectious was estimated to be 0.9% (95%CI: 0.6 to 1.2), with a lengthy prolonged isolation of 6.8 days (95% empirical CI: 1 to 8) when considering $10^5$ copies/mL as the infectiousness threshold value (*Figure 2AB*). The estimated probability of prematurely ending isolation becomes 6.6% (95%CI: 5.8 to 7.4) and 0% with a prolonged isolation of 4.8 days (95% empirical CI: −2 to 8) and 8.3 days (95% empirical CI: 6 to 10) if we consider $10^{4.5}$ and $10^{5.5}$ copies/mL as infectiousness threshold values, respectively (*Figure 2A*). To guarantee a probability lower than 5%, we estimated that patients need to be isolated for 7 days, 11 days, and 5 days for infectiousness threshold values of $10^{5.0}$, $10^{4.5}$, and $10^{5.5}$ copies/mL, respectively (*Figure 2A*). In this case, again, the length of the prolonged isolation was estimated to be substantial (*Figure 2B*): 3.8 days (95% empirical CI: −2 to 5), 5.8 days (95% empirical CI: −2 to 8), and 3.3 days (95% empirical CI: 0 to 4) for infectiousness threshold values of $10^{5.0}$, $10^{4.5}$, and $10^{5.5}$ copies/mL, respectively. In sum, to guarantee low probabilities to prematurely end the isolation and thus release patients who are still infectious, the associate cost is to have unnecessary long isolations for the majority of patients.

## Personalized approach using PCR test results

By considering two consecutive negative test results repeated at an interval of 1 day as the criterion for ending isolation, the probability of prematurely ending isolation was estimated to be 8.1% (95% CI: 7.2 to 9.0) with a negligible length of prolonged isolation of 1.2 days (95% empirical CI: −1 to 3) when considering $10^{5.0}$ copies/mL as the infectiousness threshold value (*Figure 3A*). By acting on

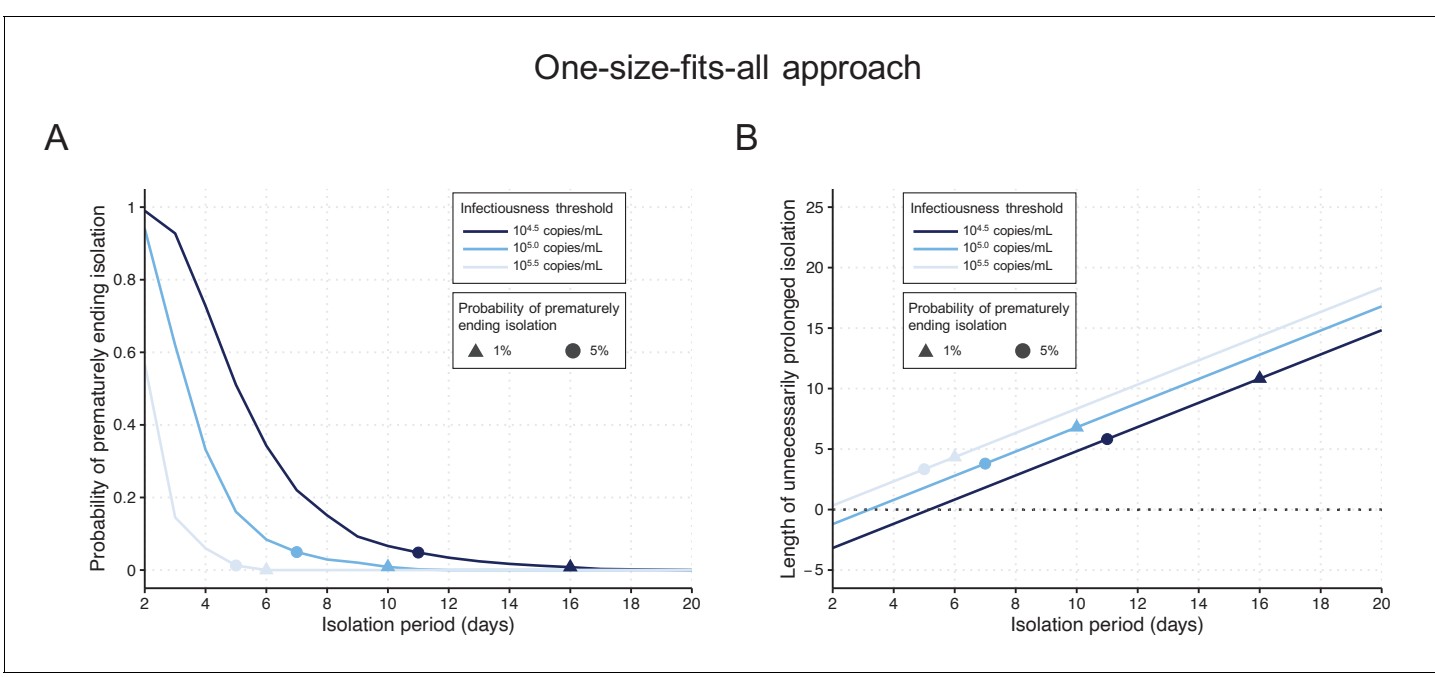

**Figure 2.** One-size-fits-all approach. (A) Probability of prematurely ending the isolation of infectious patients for different lengths of the isolation period and for different infectiousness threshold values. (B) Mean length of unnecessarily prolonged isolation for different lengths of the isolation period and for different infectiousness threshold values. Color keys and symbols apply to both panels. Note that the symbols correspond to the shortest isolation periods when the condition is met.

The online version of this article includes the following source data for figure 2:

**Source data 1.** Probability of prematurely ending isolation and mean length of unnecessarily prolonged isolation under the one-size-fits-all approach.

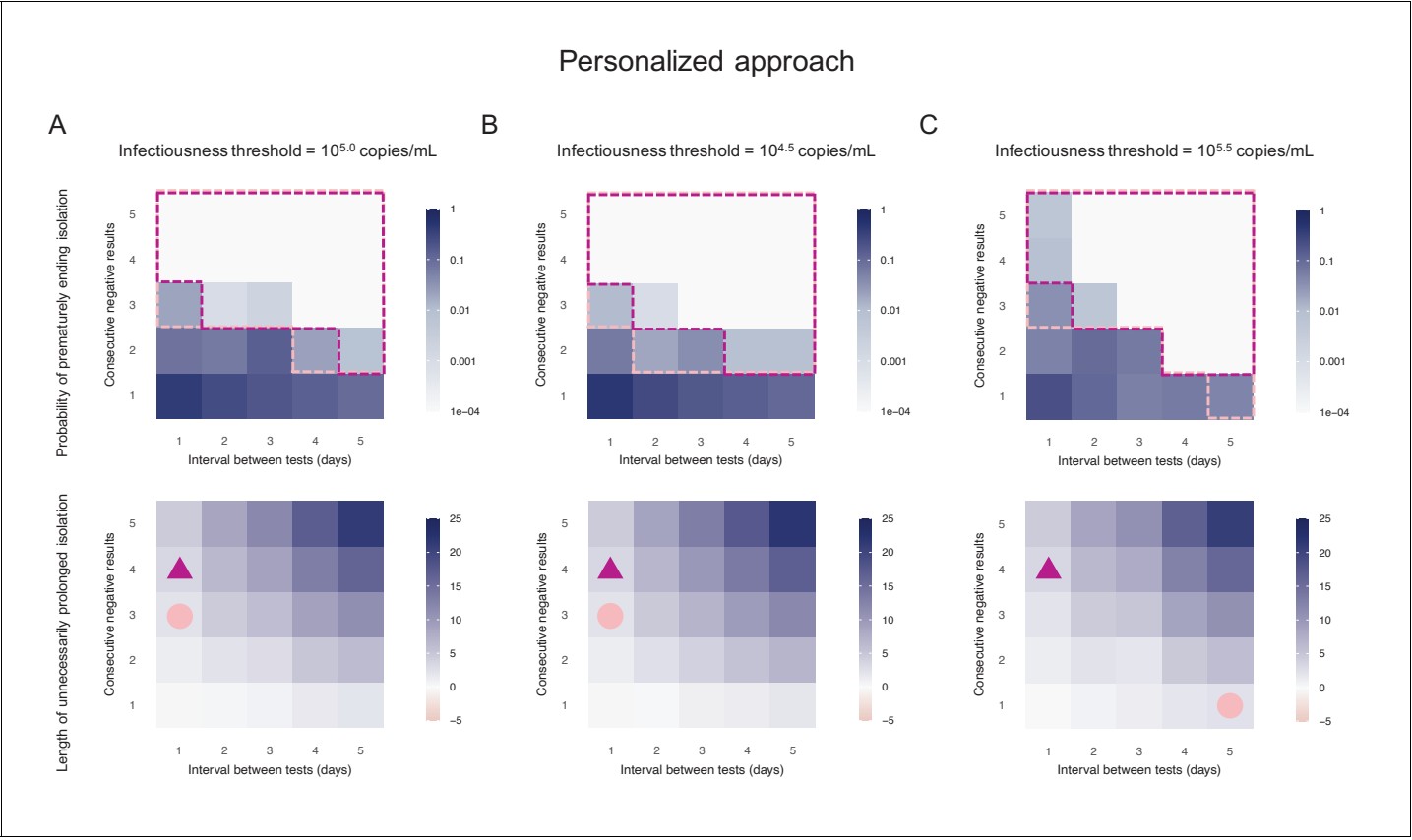

**Figure 3.** Personalized approach. (A) Probability of prematurely ending isolation (upper panels) and mean length of unnecessarily prolonged isolation (lower panels) for different values of the interval between PCR tests and the number of consecutive negative results necessary to end isolation; the infectiousness threshold value is set to $10^{5.0}$ copies/mL. The areas surrounded by purple and pink dotted lines are those with 1% or 5% or lower of risk of prematurely ending isolation of infectious patients, respectively, and the triangles and circles correspond to the conditions which realize the shortest prolonged isolation within each area. (B) Same as A, but for an infectiousness threshold value of $10^{4.5}$ copies/mL. (C) Same as A, but for an infectiousness threshold value of $10^{5.5}$ copies/mL. Color keys and symbols apply to all panels.

The online version of this article includes the following source data for figure 3:

**Source data 1.** Probability of prematurely ending isolation of infectious patients with different guidelines (with $10^{5.0}$ copies/mL as an infectiousness threshold value).

**Source data 2.** Length of unnecessarily prolonged isolation with different guidelines (with $10^{5.0}$ copies/mL as an infectiousness threshold value).

**Source data 3.** Probability of prematurely ending isolation of infectious patients with different guidelines (with $10^{4.5}$ copies/mL as an infectiousness threshold value).

**Source data 4.** Length of unnecessarily prolonged isolation with different guidelines (with $10^{4.5}$ copies/mL as an infectiousness threshold value).

**Source data 5.** Probability of prematurely ending isolation of infectious patients with different guidelines (with $10^{5.5}$ copies/mL as an infectiousness threshold value).

**Source data 6.** Length of unnecessarily prolonged isolation with different guidelines (with $10^{5.5}$ copies/mL as an infectiousness threshold value).

the testing strategy, we can control both the probability of prematurely ending isolation and the length of prolonged isolation. The probability of ending isolation of infectious patients decreased with a longer interval between testing and more consecutive negative results (the upper panel in *Figure 3A*). However, the length of prolonged isolation increased at the same time (the lower panel in *Figure 3A*). If a 5% or lower risk of prematurely ending isolation is considered, three consecutive negative test results with the tests performed every day minimizes the length of unnecessary isolation (2.3 days [95% empirical CI: 0 to 5]) (*Figure 3A*). We repeated the same analyses using different infectiousness threshold values ($10^{4.5}$ and $10^{5.5}$ copies/mL). Both the probability of prematurely ending isolation and the length of prolonged isolation were not much influenced by infectiousness

threshold values, because the viral load is directly measured in the personalized approach (*Figure 3B,C*).

## Comparison between the one-size-fits-all and the personalized approach

To highlight the differences between the one-size-fits-all and the personalized approaches, we systematically compared the two approaches by looking at the length of the prolonged isolation for a 5% or lower (*Figure 4A*) or 1% or lower (*Figure 4B*) risk of prematurely ending isolation. For the personalized approach, the best combination of the number of consecutive negative test results and the interval of tests was selected for each infectiousness threshold value. The personalized approach was not influenced by the infectiousness threshold values and yielded to shorter prolonged isolation compared with the one-size-fits-all approach. However, because the prolonged isolation for the one-size-fits-all approach was influenced by infectiousness threshold values, the difference between the one-size-fits-all and personalized approaches in prolonged isolation became smaller with higher infectiousness threshold values.

## Influence of model selection

*Figure 5* shows the length of the prolonged isolation for a 5% or lower or 1% or lower risk of prematurely ending the isolation for all the analyzed models. Regardless of the considered models, the personalized approach allows shorted length of unnecessarily isolation. Nonetheless, it is important to remark that the length of prolonged isolation is slightly different among the analyzed models. For example, under the one-size-fits-all approach, it was longer for the 'innate immune response' model as compared with the other two; this is due to larger variability in viral load especially at the late phase of the infection (*Figure 1*). Under the personalized approach, the length of prolonged isolation was longer in the baseline model as compared to the two alternative models (*Figure 1*). In

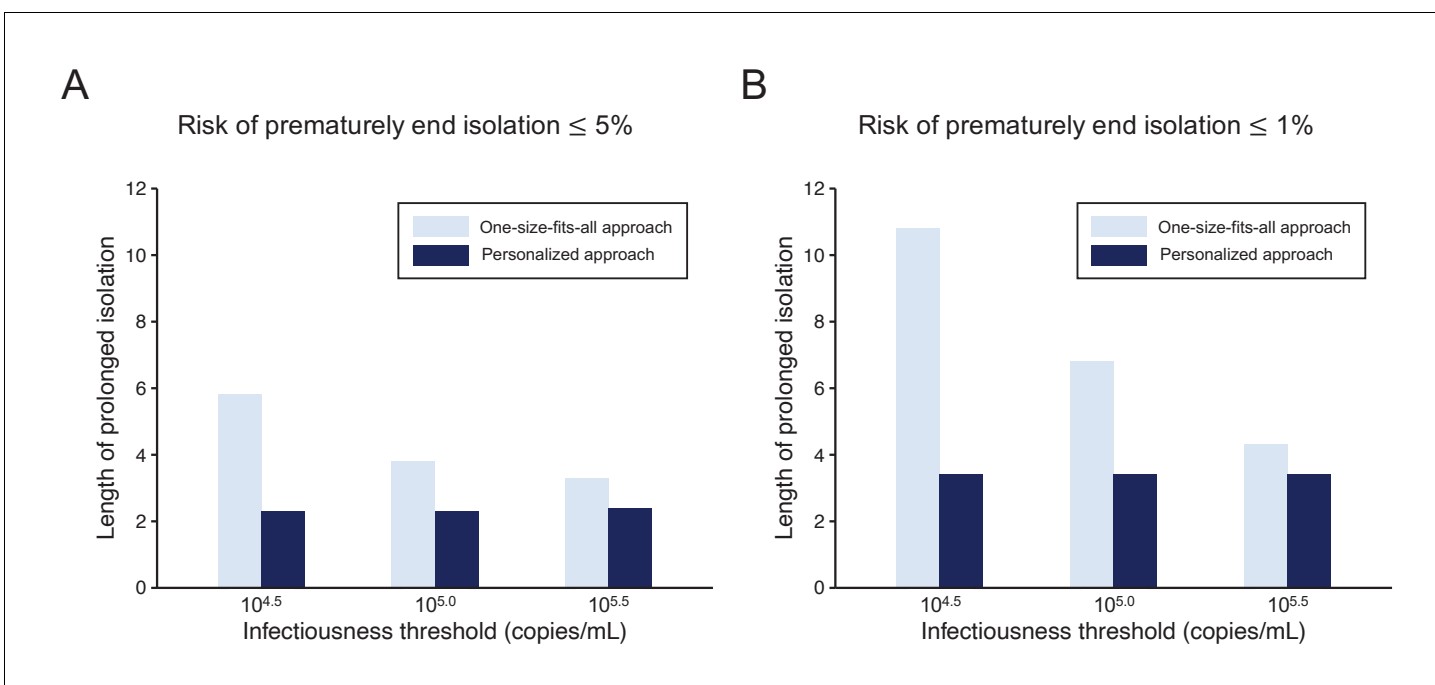

**Figure 4.** Comparison between the one-size-fits-all and the personalized approach. (**A**) Mean length of prolonged isolation for different infectiousness threshold values and for the two approaches when considering a 5% or lower risk of prematurely ending isolation. Note that for the personalized approach, the interval between PCR tests and the number of consecutive negative results necessary to end isolation were selected to minimize the duration of prolonged isolation. (**B**) Same as A, but considering a 1% or lower risk of prematurely ending isolation. Color keys apply to both panels. The online version of this article includes the following source data for figure 4:

**Source data 1.** Mean length of unnecessarily prolonged isolation (days) with different guidelines and infectiousness threshold values controlling the risk of prematurely ending isolation ≤ 5% and ≤ 1%.

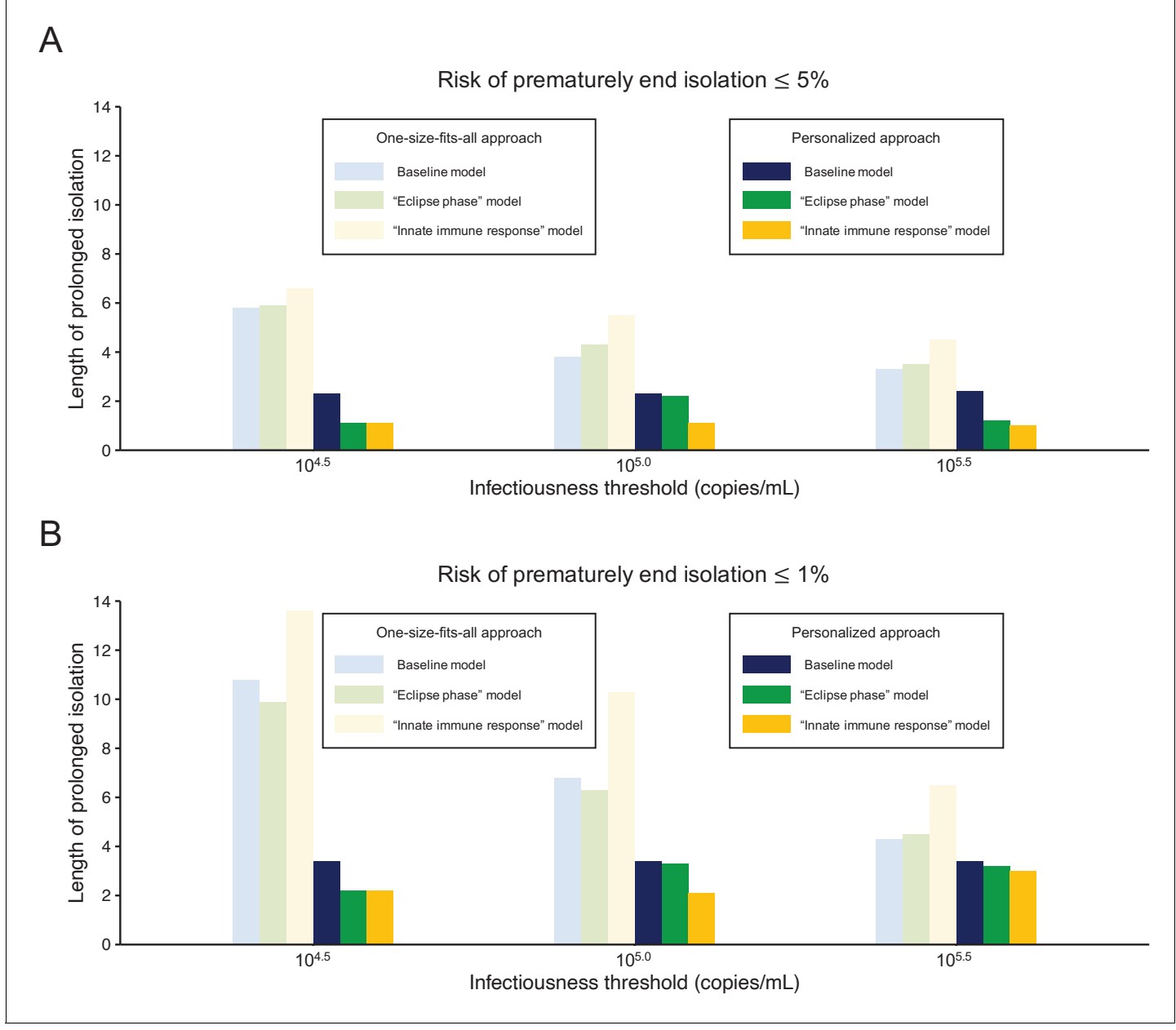

**Figure 5.** Comparison between alternative models. (**A**) Mean length of prolonged isolation for different infectiousness threshold values and for the two approaches when considering a 5% or lower risk of prematurely ending isolation and for the three analyzed models. Note that for the personalized approach, the interval between PCR tests and the number of consecutive negative results necessary to end isolation were selected to minimize the duration of prolonged isolation. (**B**) Same as A, but considering a 1% or lower risk of prematurely ending isolation. Color keys apply to both panels. The online version of this article includes the following source data for figure 5:

**Source data 1.** Mean length of unnecessarily prolonged isolation (days) with different guidelines and infectiousness threshold values controlling the risk of prematurely ending isolation ≤ 5% and ≤ 1% for the three analyzed models.

summary, by comparing the three models, we can conclude that the one-size-fits-all approach is sensitive to the variability of the viral load curve, whereas the personalized approach is sensitive to the decay speed of the viral load.

## Discussion

Guidelines for ending the isolation of COVID-19 patients that balance the risk of prematurely ending isolation with the burden of prolonged isolation are a crucial topic of discussion. Here, we propose a highly flexible modeling framework to quantify both viral dynamics and measurement errors. Using this approach, we tested alternative policies regulating the isolation of SARS-CoV-2-infected individuals by accounting for individual variability in the immune response. We estimated the probability of prematurely ending isolation and the length of unnecessarily prolonged isolation with two approaches: the one-size-fits-all approach and the personalized approach using PCR test results.

By considering a risk of 5% or lower of prematurely ending the isolation of a SARS-CoV-2-infected individual, our central estimate for the one-size-fits-all approach requires an isolation period of 7 days after symptom onset, with a prolonged isolation phase lasting about 4 days, depending on the threshold for infectiousness considered. On the other hand, the personalized approach entails a prolonged isolation phase of approximately 2 days, independently of the considered infectiousness threshold values. The better performance of the personalized approach is not surprising. In this approach, viral load is observed directly and is compared against the threshold by using PCR test results. By contrast, the one-size-fits-all approach considers only the time since symptom onset and does not refer to viral load, which has substantial interindividual variation. Further, the personalized approach can be optimized by choosing the best testing schedule (i.e. interval of testing and the number of consecutive negative test results). However, it should be noted that the personalized approach is more costly, due to the need for performing multiple PCR tests, thus entailing logistic challenges because patients need to be tested by health care professionals. The logistics of testing isolated patients is particularly challenging in Western countries, where patients not requiring hospital care are isolated in their place of residence (*European Centre for Disease Prevention and Control, 2020b*), in contrast with countries like China, where they are isolated in dedicated facilities (*Burki, 2020*). The development of PCR tests using saliva samples may help to overcome some of these challenges, promising to decrease the work burden and lower the risk of infection for health care workers (*Azzi et al., 2020*; *Tu et al., 2020*; *Wyllie et al., 2020*). Indeed, the viral load measured from saliva is comparable to or slightly higher than that from nasopharyngeal samples, which guarantees a similar level of sensitivity (*Tu et al., 2020*; *Wyllie et al., 2020*).

In this study, we used PCR tests to define the end of an isolation period in the personalized approach. PCR tests provides quantitative viral load estimates, which can be directly compared against the infectiousness threshold. Meanwhile, reverse transcription loop-mediated isothermal amplification (RT-LAMP) tests and rapid antigen tests for SARS-CoV-2 have been developed and recommended for repeated screenings, given that they are less expensive and with a shorter turnaround time than PCR tests (less than an hour vs. a day or two) (*Butler et al., 2021*; *Dao Thi et al., 2020*; *Larremore et al., 2021*; *Yang et al., 2021b*). Although these tests have lower sensitivity (the detection limit is about $10^{5.0}$ copies/mL; *Butler et al., 2021*; *Dao Thi et al., 2020*; *Miyakawa et al., 2021*; *Yang et al., 2021b*) than PCR tests, they can help mitigating SARS-CoV-2 transmission when used for population screenings (*Larremore et al., 2021*) and contact tracing (*Quilty et al., 2021*); in fact, the viral load threshold of infectiousness is considered to be higher than the detection limits of RT-LAMP tests and rapid antigen tests. Epidemiological studies are needed to assess whether isolation strategies based on RT-LAMP or rapid tests have a similar mitigation effect to those based on PCR testing.

Two guidelines for ending isolation were considered in this study. In most countries, the one-size fits-all approach is employed; however, the duration is slightly different among countries. The WHO recommends isolation for 10 days after symptom onset or a positive test for asymptomatic individuals (*World Health Organization, 2020*). The ECDC recommends isolation of 10 or 20 days for mild/moderate or severe cases, respectively, whereas for asymptomatic individuals, 10 days isolation after a positive test is recommended (*European Centre for Disease Prevention and Control, 2020a*). However, these durations actually vary from 7 to 14 days depending on each member state of the European Union (*European Comission, 2020*).

We submit that our approach can be used as a scientific backup or to adjust isolation guidelines currently in use in different countries. Nonetheless, the following limitations should be kept in mind. First, the number of samples analyzed were relatively small (30 patients), they did not cover all age groups, and pertained to symptomatic hospitalized patients only. This did not allow us to test

whether the duration of the isolation is influenced by the severity of the disease. In particular, the duration may be shorter than that predicted in this study, as the analyzed samples were composed by hospitalized patients. Guidelines considering fixed durations of the isolation depending on disease severity may be easier to implement and limit the length of unnecessary prolonged isolations. Still concerning the analyzed sample, it is important to stress that patients were infected in the early stages of the pandemic and thus likely infected by the historical SARS-CoV-2 lineages. To what extent our findings can be generalized to other categories of individuals (e.g. asymptomatic infections) and SARS-CoV-2 variants remains to be seen. The confidence intervals reported in this paper need to be cautiously interpreted as the extent and quality of relevant viral load data is unfortunately quite limited. Further, should the input data be non-representative, this could have caused a bias in our estimates of the duration of the isolation period. It is, however, important to stress that exactly in light of this scarcity of longitudinal data, model-based simulations are a powerful tool for properly integrating temporal trends in the collected data and for assessing individual variabilities. Second, we did not explicitly model the longitudinal clinical course of symptoms in SARS-CoV-2-infected individuals because of the lack of data associating the clinical course with viral load. Further research in this direction is warranted (*He et al., 2020*), especially as several countries (including the US) consider the presence or absence of symptoms among the criteria for ending isolation. Third, a deeper knowledge of the association between the viral load and the transmission risk would be a key to narrow the uncertainty surrounding the minimum viral load level that still allows SARS-CoV-2 transmission. Specifically, $10^{4.5}$, $10^{5.0}$, and $10^{5.5}$ copies/mL were used and were based on epidemiological observations of transmission events from contact tracing data (*Hu et al., 2020*; *Sun et al., 2021*). Other studies used a different perspective to approach the same research question and investigated the threshold relying on experimental virological data (i.e. culturability). For example, *van Kampen et al., 2021*; *Wölfel et al., 2020* found that the virus was culturable if the viral load is above $10^{6.0}$ copies/mL. Such uncertainty is reflected in the high variability in the results obtained for the one-size-fits-all approach; on the other hand, the personalized approach provided stable results with respect to the infectiousness threshold values. Fourth, we considered arbitrary values for the risk of prematurely ending isolation (namely, 1% or 5%). Whether such risks are acceptable depends on several factors such as the epidemiological context (e.g. the prevalence of the infection and disease burden), the aim of the adopted policies (e.g. suppression of transmission, mitigation of disease burden), propensity to take risks. Nonetheless, it is worth remarking that, if for a certain level of risk, the difference between the personalized and one-size-fits-all approaches is small, the fixed duration approach may have the advantage in terms of simplicity, cost, and resources. We also note that in the personalized approach, we used qualitative PCR test results only (i.e. whether the viral load is above or below a given threshold). The use of quantitative PCR test results may enable us to predict the optimal day to end isolation for each patient. Finally, although some of patients were tested (and isolated) before symptom onset or a few days after symptom onset, in this study, we assumed the testing starts immediately after symptom onset. As this analysis primarily focuses on the time when the viral load crosses the infectiousness threshold, we do not expect that the timing of the first test does not influence much our findings. However, starting the tests too early since isolation (or symptom onset) might be impractical and it should be determined based on operational and cost constraints. Future research could be dedicated to examining whether the starting day of testing could be defined on the basis of disease severity.

The guidelines regulating the length of isolation of COVID-19 patients require further updating following new epidemiologic and clinical knowledge, patient characteristics, and the capability of health sectors, such as test availability. Indeed, in several countries, these guidelines have been updated several times throughout the course of the pandemic (*Centers for Disease Control and Prevention, 2020a*; *European Centre for Disease Prevention and Control, 2020a*; *Public Health England, 2020*) and the emergence of new variants can spark new adjustments in the future as well. Our proposed modeling framework is very flexible and could be easily adapted to simulate the immune response and effect of antiviral therapies as well as to the study of other infectious diseases. In particular, it might prove quite relevant should new SARS-CoV-2 variants show different temporal infectiousness profiles than the historical lineage (*Davies et al., 2021*).

In conclusion, until the vaccination effort successfully suppresses the widespread circulation of SARS-CoV-2, nonpharmaceutical interventions, and patient isolation in particular, will continue to be

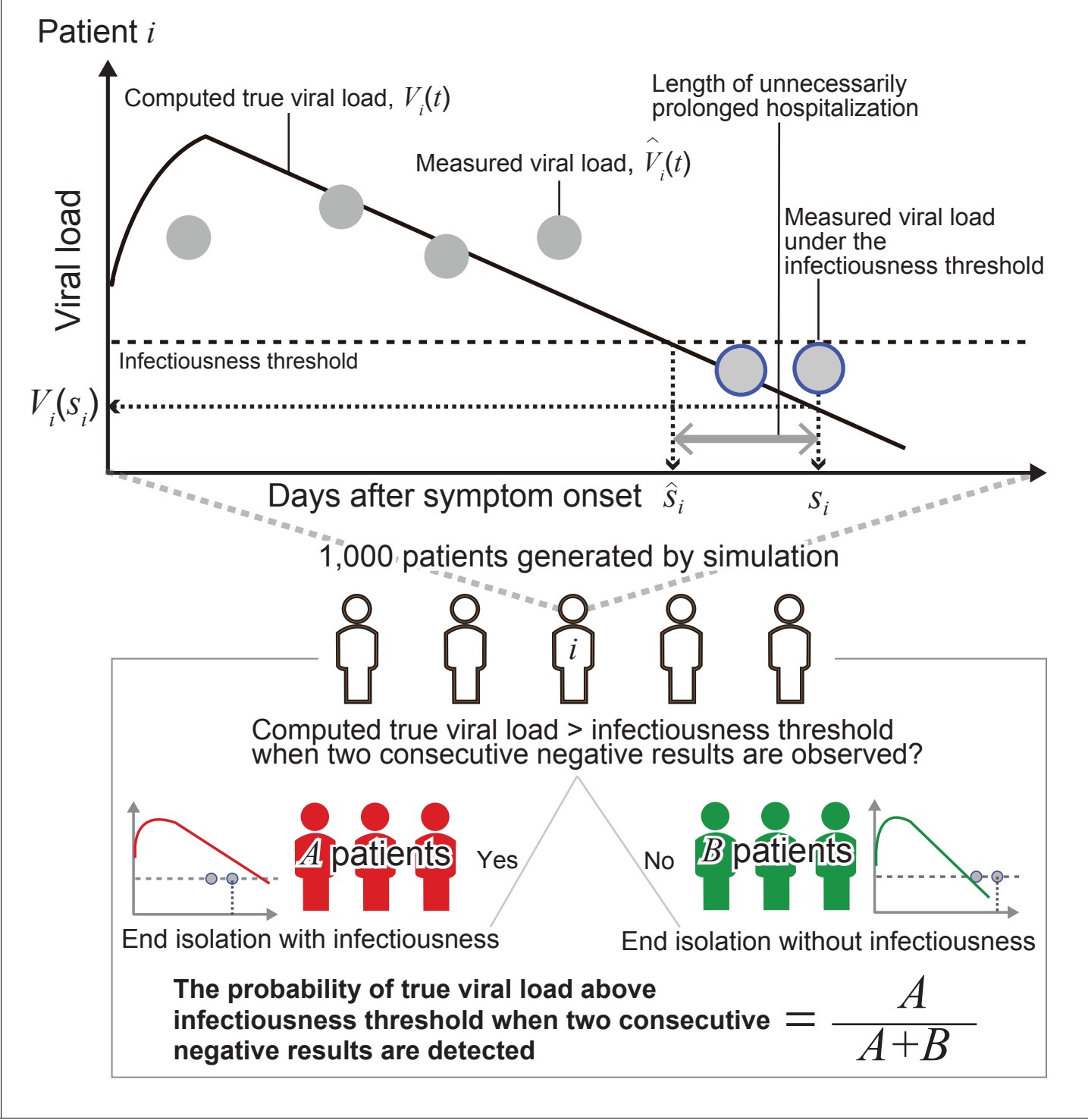

**Figure 6.** Schematic of the adopted methodology.

a primary tool for mitigating SARS-CoV-2 spread. Understanding when isolated patients may be released will thus remain a key component in the fight against COVID-19.

## Materials and methods

### Viral load data

We searched PubMed and Google Scholar for papers reporting longitudinal viral load data of COVID-19 patients. We set five inclusion criteria: (1) multiple observations of the viral load were reported per patient (if cycle thresholds were reported instead of viral load, they were transformed to viral load by using the following conversion formula [*Zou et al., 2020*]: $\log_{10}(\mathrm{Viralload[copies/mL]}) = -0.32 \times \mathrm{Ctvalues[cycles]} + 14.11$); (2) viral load was measured from upper respiratory specimens (i.e. nose, pharynx) for consistency; (3) viral load along with the time since symptom onset was reported; (4) the patients were not under antiviral treatment (antiviral therapy can directly influence the viral dynamics); and (5) patients were symptomatic (because we used the time since symptom onset as the time scale). We used the de-identified secondary data from published studies, and thus ethics approval for this study was not necessary.

### Modeling SARS-CoV-2 viral dynamics

We developed a mathematical model of SARS-CoV-2 viral dynamics (*Ikeda et al., 2016*; *Kim et al., 2020b*; *Perelson, 2002*). The model is composed of two components: (1) the ratio between the number of uninfected target cells at time $t$ and the number of uninfected target cells at time 0 ($t = 0$ corresponds to the time of symptom onset), $f(t)$; and (2) the amount of virus per unit in sample specimen (copies/mL) at time $t$, $V(t)$. $V(t)$ exponentially increases since infection, reaches a peak, and starts declining because of the depletion of target cells, which is consistent with the observed viral dynamics. Model parameters were calibrated by fitting the longitudinal data with a mixed-effect model. Details on the model and the fitting procedure are reported in Appendix 1. To account for individual variability in the viral dynamics, we simulated $V(t)$ for 1000 patients by sampling from the posterior distributions of the model parameters. To simulate the viral load measured by a PCR test, $V(t)$, we added a measurement error to $V(t)$ (see Appendix 1 for details). The model used here (baseline model) is one of the simplest models for viruses causing acute respiratory infection. Given that the biological infection process has not been fully understood yet, we believe using a simple model represents an appropriate baseline choice. Specifically in the literature of SAS-CoV-2 studies, several different models have been used including the baseline model (*Gonçalves et al., 2020*; *Goyal et al., 2020*; *Goyal et al., 2021*; *Kim et al., 2020b*). Nonetheless, to investigate to what extent the model choice affects our findings, we considered two alternative models: the 'eclipse phase' model (*Baccam et al., 2006*; *Gonçalves et al., 2020*) and the 'innate immune response' model (*Baccam et al., 2006*). The detailed description of the analyzed models is reported in Appendix 1.

### Assessment and comparison of the one-size-fits-all and the personalized approach

In the one-size-fits-all approach, we assumed isolation to end after a fixed time since symptom onset, whereas in the personalized approach using PCR tests, isolation ends after obtaining a given number of consecutive negative test results (with a given time interval between the tests). As the baseline scenario for the one-size-fits-all approach (here referred to as the 'symptom-based strategy'), we considered a fixed time of 10 days. As the baseline scenario for the personalized approach (here referred to as the 'test-based strategy'), we considered two consecutive negative test results repeated at a daily interval (in agreement with the CDC guidelines [*Centers for Disease Control and Prevention, 2020a*]). Here, we assumed the testing starts immediately after symptom onset.

Epidemiological studies based on contact tracing data suggest that infectiousness nearly disappears 8 days after symptom onset (*Hu et al., 2020*; *Sun et al., 2021*). According, to *Kim et al., 2020b*, the 97.5 percentile of the viral load 8 days after symptom onset is $10^{5.0}$ copies/mL (*Kim et al., 2021*). We thus use $10^{5.0}$ copies/mL as threshold to define whether a patient is still infectious (i.e. able to transmit the infection). All the obtained results are reported also by considering viral load threshold values of $10^{4.5}$ copies/mL and $10^{5.5}$ copies/mL as sensitivity analyses.

To evaluate the different strategies, we computed two metrics based on the simulated viral loads: the probability of prematurely ending isolation and the length of unnecessarily prolonged isolation. The probability of prematurely ending isolation is the chance that infected patients are released

from isolation while they are still infectious. The length of prolonged isolation is defined as the difference between the time at which a patient is no longer infectious and the time when her or his isolation ends. Note that when the prolonged isolation is negative, it means that the isolation period has ended when the patient is still infectious.

As sensitivity analyses, we considered the length of isolation in the range of 2 to 20 days for the one-size-fits-all approach. For the personalized approach, the frequency of testing (i.e. the interval between consecutive tests) was considered to vary between 1 and 5 days and the number of consecutive negative test results to vary between 1 and 5. Details about the performed analyses are reported in Appendix 1 and a schematic of the methodology is shown in *Figure 6*.

## Acknowledgements

This study was supported in part by Basic Science Research Program through the National Research Foundation of Korea funded by the Ministry of Education 2019R1A6A3A12031316 (to KSK); Grants-in-Aid for JSPS Scientific Research (KAKENHI) Scientific Research B 18KT0018 (to SIWAMI), 18H01139 (to SIWAMI), 16H04845 (to SIWAMI), Scientific Research S 15H05707 (to KA), Scientific Research in Innovative Areas 20H05042 (to SIWAMI), 19H04839 (to SIWAMI), 18H05103 (to SIWAMI); AMED JP20dm0307009 (to KA); AMED CREST 19gm1310002 (to SIWAMI); AMED Japan Program for Infectious Diseases Research and Infrastructure, 20wm0325007h0001, 20wm0325004s0201, 20wm0325012s0301, 20wm0325015s0301 (to SIWAMI); AMED Research Program on HIV/AIDS 19fk0410023s0101 (to SIWAMI); AMED Research Program on Emerging and Re-emerging Infectious Diseases 19fk0108156h0001, 20fk0108140s0801 and 20fk0108413s0301 (to SIWAMI); AMED Program for Basic and Clinical Research on Hepatitis 19fk0210036h0502 (to SIWAMI); AMED Program on the Innovative Development and the Application of New Drugs for Hepatitis B 19fk0310114h0103 (to SIWAMI); Moonshot R and D Grant Number JPMJMS2021 (to KA and SIWAMI) and JPMJMS2025 (to S.IWAMI); JST MIRAI (to SIWAMI); Mitsui Life Social Welfare Foundation (to SIWAMI); Shin-Nihon of Advanced Medical Research (to SIWAMI); Suzuken Memorial Foundation (to SIWAMI); Life Science Foundation of Japan (to SIWAMI); SECOM Science and Technology Foundation (to SIWAMI); The Japan Prize Foundation (to SIWAMI), and Foundation for the Fusion of Science and Technology (to SIWAMI); Taiju Life Social Welfare Foundation (to KW); Takeda Science Foundation (to KW). This research was supported through the MIDAS Coordination Center (MIDASSUGP2020-6) by a grant from the National Institute of General Medical Science (3U24GM132013-02S2) (to KE and MA). The study does not necessarily represent the views of the funding agencies listed above.

## Additional information

### Competing interests

Shingo Iwami: is an employee of Science Groove Inc. The other authors declare that no competing interests exist.

### Funding

| Funder | Grant reference number | Author |
| --- | --- | --- |
| National Research Foundation of Korea | 2019R1A6A3A12031316 | Kwang Su Kim |
| Japan Society for the Promotion of Science | 18KT0018 | Shingo Iwami |
| Japan Society for the Promotion of Science | 18H05103 | Shingo Iwami |
| Japan Society for the Promotion of Science | 18H01139 | Shingo Iwami |
| Japan Society for the Promotion of Science | 19H04839 | Shingo Iwami |
| Japan Society for the Promo- | 15H05707 | Kazuyuki Aihara |

| | | |
|---|---|---|
| tion of Science | | |
| Japan Society for the Promotion of Science | 20H05042 | Shingo Iwami |
| Japan Agency for Medical Research and Development | JP20dm0307009 | Kazuyuki Aihara |
| Japan Agency for Medical Research and Development | 19gm1310002 | Shingo Iwami |
| Japan Agency for Medical Research and Development | 20wm0325007h0001 | Shingo Iwami |
| Japan Agency for Medical Research and Development | 20wm0325004s0201 | Shingo Iwami |
| Japan Agency for Medical Research and Development | 20wm0325012s0301 | Shingo Iwami |
| Japan Agency for Medical Research and Development | 20wm0325015s0301 | Shingo Iwami |
| Japan Agency for Medical Research and Development | 19fk0410023s0101 | Shingo Iwami |
| Japan Agency for Medical Research and Development | 19fk0108156h0001 | Shingo Iwami |
| Japan Agency for Medical Research and Development | 20fk0108140s0801 | Shingo Iwami |
| Japan Agency for Medical Research and Development | 20fk0108413s0301 | Shingo Iwami |
| Japan Agency for Medical Research and Development | 19fk0210036h0502 | Shingo Iwami |
| Japan Agency for Medical Research and Development | 19fk0310114h0103 | Shingo Iwami |
| Japan Society for the Promotion of Science | 16H04845 | Shingo Iwami |
| Japan Science and Technology Agency | JPMJMS2021 | Kazuyuki Aihara |
| Japan Science and Technology Agency | JPMJMS2021 | Shingo Iwami |
| Japan Science and Technology Agency | JST MIRAI | Shingo Iwami |
| Japan Science and Technology Agency | JPMJMS2025 | Shingo Iwami |
| Mitsui Sumitomo Insurance Welfare Foundation | | Shingo Iwami |
| Advanced Medical Research Foundation | | Shingo Iwami |
| Suzuken Memorial Foundation | | Shingo Iwami |
| Life Science Foundation of Japan | | Shingo Iwami |
| Secom Science and Technology Foundation | | Shingo Iwami |
| Japan Prize Foundation | | Shingo Iwami |
| Fusion Energy Sciences | | Shingo Iwami |
| Taiju Life Social Welfare Foundation | | Koichi Watashi |
| Takeda Science Foundation | | Koichi Watashi |
| National Institute of General Medical Sciences | 3U24GM132013-02S2 | Keisuke Ejima Marco Ajelli |

The funders had no role in study design, data collection and interpretation, or the decision to submit the work for publication.

## Author contributions
Yong Dam Jeong, Data curation, Formal analysis, Investigation, Visualization, Methodology, Writing - original draft, Writing - review and editing; Keisuke Ejima, Conceptualization, Formal analysis, Supervision, Investigation, Methodology, Writing - original draft, Project administration, Writing - review and editing; Kwang Su Kim, Formal analysis, Investigation, Visualization, Methodology, Writing - original draft, Writing - review and editing; Shoya Iwanami, Formal analysis, Investigation, Methodology, Writing - original draft, Writing - review and editing; Ana I Bento, Yasuhisa Fujita, Il Hyo Jung, Koichi Watashi, Writing - original draft, Writing - review and editing; Kazuyuki Aihara, Investigation, Writing - original draft, Writing - review and editing; Taiga Miyazaki, Takaji Wakita, Conceptualization, Writing - original draft, Writing - review and editing; Shingo Iwami, Conceptualization, Supervision, Funding acquisition, Investigation, Methodology, Writing - original draft, Project administration, Writing - review and editing; Marco Ajelli, Conceptualization, Supervision, Investigation, Writing - original draft, Project administration, Writing - review and editing

## Author ORCIDs
Yong Dam Jeong (iD) https://orcid.org/0000-0002-3970-7690
Keisuke Ejima (iD) https://orcid.org/0000-0002-1185-3987
Shingo Iwami (iD) https://orcid.org/0000-0002-1780-350X

## Decision letter and Author response
Decision letter https://doi.org/10.7554/eLife.69340.sa1
Author response https://doi.org/10.7554/eLife.69340.sa2

# Additional files
## Supplementary files
- Supplementary file 1. Estimated parameters of the three models.
- Supplementary file 2. AIC and BIC of the three models.
- Transparent reporting form

## Data availability
The viral load data were directly extracted from published literature. The data are publicly available, however, we are not allowed to share the data. The sources of the data are summarized in Table 1. The data to replicate the figures are available as source data. The detailed computational process is available in Appendix 1.

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

# Appendix 1

## Data

In the study from Singapore (n=12), (*Young et al., 2020*) specimens (blood, stool, urine samples, and nasopharyngeal swabs) were collected over time for the first 2 weeks since study enrollment (all patients were hospitalized) and viral load was quantified by PCR. Viral load as measured in the nasopharyngeal swabs was used in the analysis for consistency with the other datasets. In the study from Zhuhai, China (n=8), (*Zou et al., 2020*), both nasal and throat swabs were collected, and the viral load was quantified by PCR. We used the viral load measured in the nasal swabs because the cycle threshold values were generally lower than the values for the throat swabs. In the second study from Germany (n=8), (*Wölfel et al., 2020*) viral load was measured from sputum, pharyngeal swabs, and stool collected every day. We used the data from the pharyngeal swabs for the analysis. In the study from Korea (n=2), (*Kim et al., 2020b*) upper (nasopharyngeal and oropharyngeal) and lower (sputum) respiratory specimens were collected daily or every other day after the diagnosis of infection. The viral load data measured in upper respiratory specimens were used for the analysis.

Eight cases reported from China and one case reported from Germany were excluded because their viral load was above the detection limit only two times or fewer. Five cases who received lopinavir-ritonavir treatment and one case with less than two data points from Singapore were excluded. Seven cases from Korea were excluded because they were under lopinavir-ritonavir treatment.

## A mathematical model for SARS-CoV-2 virus dynamics

### Baseline model

SARS-CoV-2 viral dynamics without antiviral treatment is described by a mathematical model previously proposed and applied in *Ikeda et al., 2016*; *Kim et al., 2020b*; *Perelson, 2002*:

$$\frac{dT(t)}{dt} = -\beta T(t)V(t), \tag{1}$$

$$\frac{dI(t)}{dt} = \beta T(t)V(t) - \delta I(t), \tag{2}$$

$$\frac{dV(t)}{dt} = pI(t) - cV(t), \tag{3}$$

where the three variables $T(t)$, $I(t)$, and $V(t)$ are the number of uninfected target cells, the number of infected target cells, and the amount of virus per unit of sample specimens (copies/mL) at time $t$, respectively. Note that time after symptom onset is used as the timescale; thus, $t=0$ is the time of symptom onset (the date on which symptoms of COVID-19 [fever, cough, and shortness of breath] first began [*Centers for Disease Control and Prevention, 2020b*]). The parameters $\beta$, $\delta$, $p$, and $c$ denote the rate constant for virus infection, the death rate of infected cells, the virus production rate, and the virus clearance rate, respectively. Because the virus clearance rate, $c$, is typically much larger than the death rate of the infected cells, $\delta$, in vivo, (*Ikeda et al., 2016*; *Martyushev et al., 2016*; *Nowak and May, 2000*) a quasi-steady state (QSS) for the amount of virus can be assumed: $dV(t)/dt = 0$. Then $I(t) = cV(t)/p$ is derived by solving *Equation 3*. Substituting this into *Equation 2*, we obtain

$$\frac{dV(t)}{dt} = \frac{p\beta}{c}T(t)V(t) - \delta V(t). \tag{4}$$

Further, we define the ratio between the number of uninfected target cells at time $t$ and the number of uninfected target cells at time 0: $f(t) = T(t)/T(0)$. The original three-dimensional system (*Equation 1-3*) is reduced to the following two-dimensional system:

$$\frac{df(t)}{dt} = -\beta f(t)V(t), \tag{5}$$

$$\frac{dV(t)}{dt} = \gamma f(t)V(t) - \delta V(t), \tag{6}$$

where $\gamma = p\beta T(0)/c$ corresponds to the maximum viral replication rate. Note that $f(t)$ is a monotonically decreasing function.

## 'Eclipse phase' model

A model considering an eclipse phase of infection, which slows viral growth, has been used to describe the virus dynamics of SARS-CoV-2 and other viruses (*Baccam et al., 2006*; *Gonçalves et al., 2020*). This model is driven by the following system of differential equations:

$$\frac{df_T(t)}{dt} = -\beta f_T(t)V(t),$$

$$\frac{df_I(t)}{dt} = \beta f_T(t)V(t) - kf_I(t),$$

$$\frac{dV(t)}{dt} = \gamma f_I(t) - \delta V(t),$$

where $f_T(t)$ is the ratio of uninfected cells at time $t$ to those at time 0, $f_I(t)$ is the ratio of infected cells in the eclipse phase and to the uninfected cells at time 0, and $V(t)$ is the concentration of virus (copies/mL). $1/k$ is the mean length of the eclipse phase. To decrease the number of parameters to be estimated, parameters $f_I(0)$ and $k$ were fixed at 0.1 and 3.0, respectively, as previously estimated for SARS-CoV-2 (*Gonçalves et al., 2020*).

## 'Innate immune response' model

A model considering an innate immune response (e.g., interferons [IFNs]) was used in previous influenza studies (*Baccam et al., 2006*). The model is regulated by the following system of differential equations:

$$\frac{df(t)}{dt} = -\frac{1}{1 + \eta F(t)}\beta f(t)V(t),$$

$$\frac{dV(t)}{dt} = \frac{1}{1 + \eta F(t)}\gamma f(t)V(t) - \delta V(t),$$

$$\frac{dF(t)}{dt} = sV(t) - \alpha F(t),$$

where $f(t)$ is the ratio of uninfected cells at time $t$ to those at time 0, $V(t)$ is the concentration of virus (copies/mL), and $F(t)$ is the concentration of IFNs produced from infected cells. $1/\eta$ means the concentration of INFs that produces a half-maximum rate constant for viral replication. $s$ and $\alpha$ are the rate constant which is proportional to secretion of INFs from infected cells and the rate of removal of IFNs, respectively. Parameters $\eta$, $s$, and $\alpha$ were assumed to decrease the number of free parameters and guarantee convergence.

## Parameter estimation with the nonlinear mixed-effect model

A nonlinear mixed-effect model was used to fit the model to the longitudinal viral load data. A mixed model was used (*Best et al., 2017*; *Gonçalves et al., 2020*) because it can capture the heterogeneity in viral dynamics. Both a fixed effect (same among individuals) and a random effect (different between individuals) in each parameter are considered. Specifically, the parameter for patient $k$, $\vartheta_k (= \vartheta \times e^{\pi_k})$ is a product of $\vartheta$ (a fixed effect) and $e^{\pi_k}$ (a random effect), where $\pi_k$ follows the normal distribution: $N(0, \Omega)$. Fixed effects and random effects were estimated using the stochastic approximation Expectation/Maximization (SAEM) algorithm and empirical Bayes method, respectively. To account for the left-censoring problem (i.e., when the viral load is under the detection limit, or tests negative), the likelihood function was developed using a left-truncated Gaussian distribution assuming such data are in the censoring interval (0 to the detection limit) (*Samson et al., 2006*). Finally, we fit the normal distribution ($N(0, \sigma)$) to the residuals (i.e., difference between true viral load

and measured viral load) to quantify the measurement error used in the subsequent simulation. Fitting was performed on MONOLIX 2019R2 (http://www.lixoft.com) (*Traynard et al., 2020*).

## Simulation of viral dynamics

We randomly resampled the parameter set (i.e., $\beta$, $\gamma$, $\delta$, and $V(0)$) from the estimated distribution and ran the model. We assumed viral load obtained by running the model over time, $V(t)$, is true (or expected) viral load. However, the viral load quantified by the PCR test is influenced by a measurement error. Thus, we added the measurement error to the true viral load and obtained measured viral load, $\widehat{V}(t)$: $\widehat{V}(t) = V(t) + \varepsilon, \varepsilon \sim N(0, \sigma)$. The variance of the error, $\sigma^2$, was estimated in the fitting process (see the previous section). We assumed the error is independent and identically distributed (i.e., the errors are not correlated among patients or among multiple measurements from the same patients).

## Probability of prematurely ending isolation and the length of prolonged isolation for the personalized approach

Although the interval of tests and the consecutive negative results necessary to end isolation can vary, for the purpose of illustration in the following explanation, we assumed that the tests were performed every day and that two consecutive negative results were necessary.

By running the model with parameter sets resampled from the estimated distributions, we obtain true viral load, $V(t)$ (thick black lines in *Figure 6*), and measured viral load, $\widehat{V}(t)$ (gray dots in *Figure 6*) over time since symptom onset for 1000 'virtual patients'. Negative results correspond to the measured viral load below an infectiousness threshold value (gray dots circled in blue in *Figure 6*). We denoted the timing of the second negative results as $s_i$ and the day when the true viral load drops below the infectiousness threshold value as $s_i$ for patient $i$.

Then, we computed the probability that true viral load is above the detection limit when the second consecutive negative result is obtained:

$$p = \sum_{i=1}^{1000} I(V_i(s_i) > \text{infectiousness threshold value})/1000$$

where $I$ is the identity function. In the upper panel in *Figure 6*, the true viral load is below the detection limit when the second consecutive negative result is observed. The length of unnecessarily prolonged isolation for patient $i$ was defined as the difference between the time of ending isolation, $s_i$, and the time that the true viral load drops below the infectiousness threshold value, $\hat{s}_i$: $s_i - \hat{s}_i$. The 95% confidence intervals (CIs) of the probability of prematurely ending isolation, $p$, was computed assuming a binominal distribution: $p \pm 1.96\sqrt{p(1-p)/1000}$.

