## [Decision Letter]

**Acceptance summary:**

This paper uses a simulation approach to demonstrate that a personalized viral load based testing approach has the potential to limit the duration of unnecessary isolation among infected people while not increasing the risk of releasing an infectious person. This work could influence policies regarding duration of isolation is hospitals and at home.

**Decision letter after peer review:**

Thank you for submitting your article "Revisiting the guidelines for ending isolation for COVID-19 patients" for consideration by *eLife*. Your article has been reviewed by 2 peer reviewers, including Joshua T Schiffer as the Reviewing Editor and Reviewer #1, and the evaluation has been overseen by a Senior Editor.

As is customary in *eLife*, the reviewers have discussed their critiques with one another and with the editors. What follows below is the Reviewing Editor's edited compilation of the essential and ancillary points provided by reviewers in their critiques and in their interaction post-review. Please submit a revised version that addresses these concerns directly.

Essential Revisions:

1) Please enhance the literature review to discuss the relevance of this paper in light of new, less expensive testing strategies including rapid antigen testing. Please also discuss more specifically how different countries around the world are differing in terms of isolation policies.

2) Please test different intra-host models against the data using residuals and AIC and see if different models alter conclusions about optimal isolation guidelines in anyway.

*Reviewer #1:*

1) Overall, the paper is under referenced and state of the art testing approaches are not described in sufficient detail in the intro or discussion. A brief review of national practices that incorporate one strategy or the other to inform the reader of the current standard would be helpful to highlight the importance of the work.

2) There is no mention of the use of antigen tests which are less sensitive for viral RNA but more specific for infectious virus. These are far less expensive, have far less turnaround and are now widely used in many countries. Antigen tests should be contextualized given the results of this modeling. Saliva testing is also widespread in many places. Modeling of both approaches is available in the literature and should be cited.

3) Several groups have published intra-host viral dynamic models (the Guedj and Schiffer groups) with slightly different mechanistic assumptions. These models should be discussed and in particular it should be mentioned whether their slightly different structures could alter the paper's conclusions. Similarly, a couple of research groups have made estimates of viral load thresholds required for transmission and these should be referenced as well.

4) The model is fit to very little data (as very little is available) and while the posterior sampling method to generate 1000 in silico patients is reasonable, it is no substitute for real data. The authors should acknowledge that confidence intervals reported in the paper are quite speculative in the sense that the extent and quality of relevant viral load data for intra-host modeling is unfortunately quite limited. If the model is misclassified based on non-representative input data, then estimations about duration of isolation could be biased.

5) For the test-based strategy, are there any assumptions about when the first test might occur? Can this only occur after development of symptoms or after a certain number of days of infection? This section of the methods should provide more detail.

*Reviewer #2:*

Jeong et al., examined the important question of the guidelines for ending the isolation of COVID-19 patients. Two types of guidelines are commonly used: 1) A fixed duration (10 days or 2 weeks) of isolation following the development of symptoms, which the authors call 'one-size-fits-all'. 2) Two successive RT/PCR negative test results separated by 24 hours for ending isolation, which the authors term 'personalized'. In the former, a long duration would lead to unnecessarily long isolation periods, whereas a short duration may end up releasing individuals still able to transmit the disease. The latter avoids these pitfalls, but requires multiple tests, increasing costs and the burden on healthcare staff. To identify which of these strategies is better, the authors develop a mathematical model of within-host SARS-CoV-2 dynamics and apply it to data of viral load changes post infection from untreated patients. Using the parameters estimated, they create a pool of 1000 virtual patients and simulate dynamics in these patients and assess the consequences of the two isolation ending approaches by calculating the probability that a patient released is still able to transmit and the excess or unwarranted duration of isolation. They find that in general the personalized approach fares better on both metrics.

The question is important and timely given the raging COVID-19 pandemic. The conceptual approach developed is novel and is also likely to be applicable beyond the current pandemic. The application of the approach and the resulting inferences drawn, however, need stronger justification. My reasons are below.

1. In the personalized approach, where RT/PCR tests are used, the chance that a person who is infectious is declared non-infectious (or vice versa) is due to measurement error. This error is estimated in the study as the variance of the normal distribution fit to the residuals of the best-fits of the mathematical model to the patient data of viral load changes (see lines 464 and 473). The error is thus strongly dependent on the model. One could use a model with more parameters and obtain a 'better' fit to the data, with smaller residuals, which could then presumably change the inferences above. Indeed, many other models have been developed to describe SARS-CoV-2 dynamics and have been applied to some of the datasets the present study has used.

2. A second concern, which the authors too recognize, is that the data used is all from hospitalized patients, which may not be representative of the vast fraction of infected individuals undergoing (home or institutional, but not hospital) isolation following mild/moderate symptoms. The required durations of isolation may then be even shorter than predicted. Would guidelines that account for this heterogeneity in disease severity be easier to implement? In other words, individuals could be categorized into disease severity classes (say asymptomatic, mild, moderate, or severe) and have fixed but different durations of isolation for each class. For personalized treatments, one could still use these categories to decide when to start measurements. In the present study, when to start measurements in the personalized approach is not mentioned and it appears that measurements are assumed to be made daily from the time of isolation, which may be unnecessary and impractical.

3. Finally, the two approaches are compared at 5% and 1% probabilities of ending isolation prematurely (Figure 4). While 1% appears small, whether it is small enough from an epidemiological perspective remains to be addressed. In other words, whether 1% 'leakage' of infectious individuals from isolation is tolerable would depend on the setting, particularly, the population density and the propensity for risky behavior. If at an epidemiologically identified threshold, the difference between the two approaches is small, the fixed duration approach may have the advantage of simplicity and of doing away with additional tests.

1. One way to address comment 1 above could be to compare alternative models and identify the best model based on estimates of AIC, or other such metrics, and use it to estimate the measurement error. Alternatively, if experimental uncertainties in RT/PCR measurements are known, using them instead of the variance of the normal distribution fit to residuals could provide an independent verification of the inferences.

2. The authors should show fits of the model to the data and list the model parameter values estimated; this would help appreciate the inferences drawn better.

---

## [Author Response]

Essential Revisions:1) Please enhance the literature review to discuss the relevance of this paper in light of new, less expensive testing strategies including rapid antigen testing. Please also discuss more specifically how different countries around the world are differing in terms of isolation policies.

We thank the editors and reviewers for this comment and fully agree about the importance of these new technologies. RT-LAMP tests and rapid antigen tests have been developed and recommended for repeated screening, given that they are less expensive and test results are available faster than PCR tests (10-15 mins vs. a few hours) regardless of their lower sensitivity. We add the following paragraph to discuss the current use of these tests and their application in the context of isolation strategies (Page 9 Line 195-206):

“In this study, we used PCR tests to define the end of an isolation period in the personalized approach. PCR tests provide quantitative viral load estimates, which can be directly compared against the infectiousness threshold. Meanwhile, reverse transcription loop-mediated isothermal amplification (RT-LAMP) tests and rapid antigen tests for SARS-CoV-2 have been developed and recommended for repeated screenings, given that they are less expensive and with a shorter turnaround time than PCR tests (less than an hour vs. a day or two) (Butler et al., 2021; Dao Thi et al., 2020; Larremore et al., 2021; Yang et al., 2021). Although these tests have lower sensitivity (the detection limit is about 10^5.0^ copies/mL; Butler et al., 2021; Dao Thi et al., 2020; Miyakawa et al., 2021; Yang et al., 2021) than PCR tests, they can help mitigating SARS-CoV-2 transmission when used for population screenings (Larremore et al., 2021) and contact tracing (Quilty et al., 2021); in fact, the viral load threshold of infectiousness is considered to be higher than the detection limits of RT-LAMP tests and rapid antigen tests. Epidemiological studies are needed to assess whether isolation strategies based on RT-LAMP or rapid tests have a similar mitigation effect to those based on PCR testing.”

Moreover, as suggested, we have added a more comprehensive review of isolation policies in different countries (Page 9 Line 207-213):

“Two guidelines for ending isolation were considered in this study. In most countries, the one-size-fits-all approach is employed; however, the duration is slightly different among countries. The WHO recommends isolation for 10 days after symptom onset or a positive test for asymptomatic individuals (World Health Organization, 2020). The ECDC recommends isolation of 10 or 20 days for mild/moderate or severe cases, respectively, while for asymptomatic individuals, 10 days isolation after a positive test is recommended (European Centre for Disease Prevention and Control, 2020). However, these durations actually vary from 7 to 14 days depending on each member state of the European Union (European Comission, 2020).”

2) Please test different intra-host models against the data using residuals and AIC and see if different models alter conclusions about optimal isolation guidelines in anyway.

The model we used in the study (baseline model) is one of the simplest models. As the editors and the reviewers suggested, the model can be extended to incorporate other factors. Following their suggestion, we have added two alternative models, both of which are still universally used for viruses causing acute respiratory infection.

1) “Eclipse phase” model

A model considering an eclipse phase of infection, which slows viral growth, has been used to describe the virus dynamics of SARS-CoV-2 and other viruses (Baccam, Beauchemin, Macken, Hayden, and Perelson, 2006; Gonçalves et al., 2020). This model is driven by the following system of differential equations:dfT(t)dt=−βfT(t)V(t),dfI(t)dt=βfT(t)V(t)−kfI(t),dV(t)dt=γfI(t)−δV(t),where fT(t) is the ratio of uninfected cells at time t to those at time 0, fI(t) is the ratio of infected cells in the eclipse phase to the uninfected cells at time 0, and V(t) is the concentration of virus (copies/mL). 1/k is the mean length of the eclipse phase. To decrease the number of parameters to be estimated, parametrs fI(0) and k were fixed at 0.1 and 3.0, respectively, as previously estimated for SARS-CoV-2 (Gonçalves et al., 2020).

2) “Innate immune response” model

A model considering an innate immune response (e.g., interferons [IFNs]) was used in previous influenza studies (Baccam et al., 2006). The model is regulated by the following system of differential equations:df(t)dt=−11+ηF(t)βf(t)V(t),dV(t)dt=11+ηF(t)γf(t)V(t)−δV(t),dF(t)dt=sV(t)−αF(t),where f(t) is the ratio of uninfected cells at time t to those at time 0, V(t) is the concentration of virus (copies/mL), and F(t) is the concentration of IFNs produced from infected cells. 1/η is the concentration of INFs that produces a half-maximum rate constant for viral replication. s and α are the rate constant which is proportional to secretion of INFs from infected cells and the rate of removal of IFNs, respectively.

In the revised version of the manuscript these two new models are presented alongside the baseline model and compared based on the Akaike information criterion (AIC) and the Bayesian information criterion (BIC). The three models provided very consistent results, which are summarized in Supplementary File 2 and Figure 1.

Next, to further investigate the sensitivity of our findings to the selected model, we performed the same simulations presented in the original manuscript for the baseline model also for the calibrated “eclipse phase” model and “innate immune response” model. Overall, all the three models suggest that the personalized approach allows shorted length of unnecessarily isolation and provide quantitatively similar estimates. Together with a new figure (Figure 5), the following section has been added in the main text (Page 8, Line 156-165):

“Influence of model selection Figure 5 shows the length of the prolonged isolation for a 5% or lower or 1% or lower risk of prematurely ending the isolation for all the analyzed models. Regardless of the considered models, the personalized approach allows shorted length of unnecessarily isolation. Nonetheless, it is important to remark that the length of prolonged isolation is slightly different among the analyzed models. For example, under the one-size-fits-all approach, it was longer for the “innate immune response” model as compared with the other two; this is due to larger variability in viral load especially at the late phase of the infection (Figure 1). Under the personalized approach, the length of prolonged isolation was longer in the baseline model as compared to the two alternative models (Figure 1). In summary, by comparing the three models, we can conclude that the one-size-fits-all approach is sensitive to the variability of the viral load curve, whereas the personalized approach is sensitive to the decay speed of the viral load.”

We would like to thank once again the editors and reviewers for this comment as we believe that comparing alternative models remarkably strengthens our analysis and supports the robustness of our results.

Reviewer #1:1) Overall, the paper is under referenced and state of the art testing approaches are not described in sufficient detail in the intro or discussion. A brief review of national practices that incorporate one strategy or the other to inform the reader of the current standard would be helpful to highlight the importance of the work.

We apologize for the lack of sufficient discussion and agree about the relevance of providing the reader with a better context for our analysis. We have revised the Discussion to include an overview of the isolation policies recommended by the WHO and ECDC (in addition to those recommended by the CDC) (Page 9, Lines 207-213):

“Two guidelines for ending isolation were considered in this study. In most countries, the one-size-fits-all approach is employed; however, the duration is slightly different among countries. The WHO recommends isolation for 10 days after symptom onset or a positive test for asymptomatic individuals (World Health Organization, 2020). The ECDC recommends isolation of 10 or 20 days for mild/moderate or severe cases, respectively, while for asymptomatic individuals, 10 days isolation after a positive test is recommended (European Centre for Disease Prevention and Control, 2020). However, these durations actually vary from 7 to 14 days depending on each member state of the European Union (European Comission, 2020).”

2) There is no mention of the use of antigen tests which are less sensitive for viral RNA but more specific for infectious virus. These are far less expensive, have far less turnaround and are now widely used in many countries. Antigen tests should be contextualized given the results of this modeling. Saliva testing is also widespread in many places. Modeling of both approaches is available in the literature and should be cited.

We apologize for the lack of discussion about antigen tests. We agree with reviewer about their importance, which we now acknowledge in the revised version of our manuscript (Page 9, Line 195-206). For further details about antigen testing, we refer the reviewer to the response above.

Regarding saliva testing, we have added the following paragraph to acknowledge its relevance and recognize the literature on the topic (Page 9, Line 189-194):

“The development of PCR tests using saliva samples may help to overcome some of these challenges, promising to decrease the work burden and lower the risk of infection for health care workers (Azzi et al., 2020; Tu et al., 2020; Wyllie et al., 2020). Indeed, the viral load measured from saliva is comparable to or slightly higher that from nasopharyngeal samples, which guarantees a similar level of sensitivity (Tu et al., 2020; Wyllie et al., 2020).”

3) Several groups have published intra-host viral dynamic models (the Guedj and Schiffer groups) with slightly different mechanistic assumptions. These models should be discussed and in particular it should be mentioned whether their slightly different structures could alter the paper's conclusions. Similarly, a couple of research groups have made estimates of viral load thresholds required for transmission and these should be referenced as well.

As suggested, we have now included two alternative models in our analysis. The results of the two additional models are in overall agreement with those obtained with the model presented in the originally submitted manuscript, thus reinforcing our findings. We refer the reviewer to the response above for details.

Regarding the estimates of the viral load thresholds required for transmission, we have relied on epidemiological evidence of transmission as derived from contact tracing data. Nonetheless, we agree with the reviewer that there exists an important body of literature about the connection between SARS-CoV-2 transmission and viral load. We have added the following paragraph to acknowledge the relevance of these studies (Page 10, Line 233-240):

“Third, a deeper knowledge of the association between the viral load and the transmission risk would be a key to narrow the uncertainty surrounding the minimum viral load level that still allows SARS-CoV-2 transmission. Specifically, 104.5, 105.0, and 105.5 copies/mL were used and were based on epidemiological observations of transmission events from contact tracing data (He et al., 2020). Other studies used a different perspective to approach the same research question and investigated the threshold relying on experimental virological data (i.e., capturability). For example, Wölfel et al. and van Kampen et al. found that the virus was culturable if the viral load is above 106.0 copies/mL (van Kampen et al., 2021; Wölfel et al., 2020).”

4) The model is fit to very little data (as very little is available) and while the posterior sampling method to generate 1000 in silico patients is reasonable, it is no substitute for real data. The authors should acknowledge that confidence intervals reported in the paper are quite speculative in the sense that the extent and quality of relevant viral load data for intra-host modeling is unfortunately quite limited. If the model is misclassified based on non-representative input data, then estimations about duration of isolation could be biased.

We fully agree with the reviewer that our analysis suffers from the limitations mentioned by the reviewer and we apologize for not having stressed them enough in the originally submitted version of the manuscript. We have revised the discussion as follow (Page 10, Line 224-227):

“The confidence intervals reported in this paper need to be cautiously interpreted as the extent and quality of relevant viral load data is unfortunately quite limited. Further, should the input data be non-representative, this could have caused a bias in our estimates of the duration of the isolation period.”

5) For the test-based strategy, are there any assumptions about when the first test might occur? Can this only occur after development of symptoms or after a certain number of days of infection? This section of the methods should provide more detail.

We apologize for the lack of detail. We assumed that the test is conducted (and the isolation starts) immediately after symptom onset. This has been clarified in the Methods (Page 13, Line 311-312).

Moreover, we have added the following paragraph in the Discussion to acknowledge the limits of our choice (Page 11, Line 250-256):

“Finally, although some of patients are tested (and isolated) before symptom onset or a few days after symptom onset, in this study, we assumed the testing starts immediately after symptom onset. As this analysis primarily focuses on the time when the viral load crosses the infectiousness threshold, we do not expect that the timing of the first test does not influence much our findings. However, starting the tests too early since isolation (or symptom onset) might be impractical and it should be determined based on operational and cost constraints. Future research could be dedicated to examining whether the starting day of testing could be defined on the basis of disease severity.”

Reviewer #2:Jeong et al., examined the important question of the guidelines for ending the isolation of COVID-19 patients. Two types of guidelines are commonly used: 1) A fixed duration (10 days or 2 weeks) of isolation following the development of symptoms, which the authors call 'one-size-fits-all'. 2) Two successive RT/PCR negative test results separated by 24 hours for ending isolation, which the authors term 'personalized'. In the former, a long duration would lead to unnecessarily long isolation periods, whereas a short duration may end up releasing individuals still able to transmit the disease. The latter avoids these pitfalls, but requires multiple tests, increasing costs and the burden on healthcare staff. To identify which of these strategies is better, the authors develop a mathematical model of within-host SARS-CoV-2 dynamics and apply it to data of viral load changes post infection from untreated patients. Using the parameters estimated, they create a pool of 1000 virtual patients and simulate dynamics in these patients and assess the consequences of the two isolation ending approaches by calculating the probability that a patient released is still able to transmit and the excess or unwarranted duration of isolation. They find that in general the personalized approach fares better on both metrics.The question is important and timely given the raging COVID-19 pandemic. The conceptual approach developed is novel and is also likely to be applicable beyond the current pandemic. The application of the approach and the resulting inferences drawn, however, need stronger justification. My reasons are below.

We would like to thank the reviewer for the careful evaluation of our manuscript and her/his constructive input. We are also glad that the reviewer found our research question “important and timely” and appreciated our modeling approach.

1. In the personalized approach, where RT/PCR tests are used, the chance that a person who is infectious is declared non-infectious (or vice versa) is due to measurement error. This error is estimated in the study as the variance of the normal distribution fit to the residuals of the best-fits of the mathematical model to the patient data of viral load changes (see lines 464 and 473). The error is thus strongly dependent on the model. One could use a model with more parameters and obtain a 'better' fit to the data, with smaller residuals, which could then presumably change the inferences above. Indeed, many other models have been developed to describe SARS-CoV-2 dynamics and have been applied to some of the datasets the present study has used.

We would like to thank the reviewer for this comment that allowed us to strengthen our analysis. As suggested, we have now considered two additional models (taken from the literature). The results obtained with these two additional models are in overall agreement with those obtained in the original analysis, although some quantitative differences do exist, especially in the description of the late phase of the infection.

2. A second concern, which the authors too recognize, is that the data used is all from hospitalized patients, which may not be representative of the vast fraction of infected individuals undergoing (home or institutional, but not hospital) isolation following mild/moderate symptoms. The required durations of isolation may then be even shorter than predicted. Would guidelines that account for this heterogeneity in disease severity be easier to implement? In other words, individuals could be categorized into disease severity classes (say asymptomatic, mild, moderate, or severe) and have fixed but different durations of isolation for each class. For personalized treatments, one could still use these categories to decide when to start measurements. In the present study, when to start measurements in the personalized approach is not mentioned and it appears that measurements are assumed to be made daily from the time of isolation, which may be unnecessary and impractical.

We echo the reviewer about the possibility that guidelines accounting for disease severity may represent a viable option. Unfortunately, as discussed in the originally submitted version of our manuscript and also recognized by the reviewer, the data (un)availability prevented us to test this hypothesis. We have added the following paragraph to discuss this point, which is still open for future studies (Page 10, Line 215-221):

“First, the number of samples analyzed were relatively small (30 patients), they did not cover all age groups, and pertained to symptomatic hospitalized patients only. This did not allow us to test whether the duration of the isolation is influenced by the severity of the disease. In particular, the duration may be shorter than that predicted in this study, as the analyzed samples were composed by hospitalized patients. Guidelines considering fixed durations of the isolation depending on disease severity may be easier to implement and limit the length of unnecessary prolonged isolations.”

Regarding the personalized approach, we assumed that the test is started immediately after symptom onset and we apologize for the omission of this detail. We have now specified it in the text (Page 13, Line 311-312). We also agree with the reviewer’s suggestion that the start of the testing for isolated patients may be optimized by taking into account disease severity. We have added the following paragraph to discuss this interesting topic (Page 11, Line 250-256):

“Finally, although some of patients were tested (and isolated) before symptom onset or a few days after symptom onset, in this study, we assumed the testing starts immediately after symptom onset. As this analysis primarily focuses on the time when the viral load crosses the infectiousness threshold, we do not expect that the timing of the first test does not influence much our findings. However, starting the tests too early since isolation (or symptom onset) might be impractical and it may be determined based on operational and cost constraints. Future research could be dedicated to examining whether the starting day of testing could be defined on the basis of disease severity.”

3. Finally, the two approaches are compared at 5% and 1% probabilities of ending isolation prematurely (Figure 4). While 1% appears small, whether it is small enough from an epidemiological perspective remains to be addressed. In other words, whether 1% 'leakage' of infectious individuals from isolation is tolerable would depend on the setting, particularly, the population density and the propensity for risky behavior. If at an epidemiologically identified threshold, the difference between the two approaches is small, the fixed duration approach may have the advantage of simplicity and of doing away with additional tests.

We apologize for the lack of detail. As correctly pointed out by the reviewer, 1% and 5% probabilities to prematurely ending the isolation were fully arbitrary choices made to provide two illustrative examples. Whether these risks are considered to be acceptable depends on several factors and political choices. We have added the following paragraph to clarify this key point/study limitation and acknowledge that the one-size-fits-all approach may be the best choice if for a predetermined risk level the two approaches show similar results (Page 11, Line 242-248):

“Fourth, we considered arbitrary values for the risk of prematurely ending isolation (namely, 1 or 5%). Whether such risks are acceptable depends on several factors such as the epidemiological context (e.g., the prevalence of the infection and disease burden), the aim of the adopted policies (e.g., suppression of transmission, mitigation of disease burden), propensity to take risks. Nonetheless, it is worth remarking that, if for a certain level of risk, the difference between the personalized and one-size-fits-all approaches is small, the fixed duration approach may have the advantage in terms of simplicity, cost, and resources.”

1. One way to address comment 1 above could be to compare alternative models and identify the best model based on estimates of AIC, or other such metrics, and use it to estimate the measurement error. Alternatively, if experimental uncertainties in RT/PCR measurements are known, using them instead of the variance of the normal distribution fit to residuals could provide an independent verification of the inferences.

Thank you for this comment. As suggested by the reviewer, we have added two alternative models and compared them using both AIC and BIC, finding little difference between them.

2. The authors should show fits of the model to the data and list the model parameter values estimated; this would help appreciate the inferences drawn better.

We apologize for these omissions. In the revised version, we have added the individual fitting results and the estimated parameter values in the Supplementary Information (Figure 1—figure supplement 1 and Supplementary File 1).

References

Azzi, L., Carcano, G., Gianfagna, F., Grossi, P., Gasperina, D. D., Genoni, A.,... Baj, A. (2020). Saliva is a reliable tool to detect SARS-CoV-2. *Journal of Infection, 81*(1), e45-e50. doi:10.1016/j.jinf.2020.04.005

Baccam, P., Beauchemin, C., Macken, C. A., Hayden, F. G., & Perelson, A. S. (2006). Kinetics of Influenza A Virus Infection in Humans. *Journal of Virology, 80*(15), 7590-7599. doi:10.1128/JVI.01623-05

Butler, D., Mozsary, C., Meydan, C., Foox, J., Rosiene, J., Shaiber, A.,... Mason, C. E. (2021). Shotgun transcriptome, spatial omics, and isothermal profiling of SARS-CoV-2 infection reveals unique host responses, viral diversification, and drug interactions. *Nature Communications, 12*(1), 1660. doi:10.1038/s41467-021-21361-7

Dao Thi, V. L., Herbst, K., Boerner, K., Meurer, M., Kremer, L. P., Kirrmaier, D.,... Anders, S. (2020). A colorimetric RT-LAMP assay and LAMP-sequencing for detecting SARS-CoV-2 RNA in clinical samples. *Science Translational Medicine, 12*(556), eabc7075. doi:10.1126/scitranslmed.abc7075

European Centre for Disease Prevention and Control. (2020). Guidance for discharge and ending isolation of people with COVID-19. Retrieved from https://www.ecdc.europa.eu/en/publications-data/covid-19-guidance-discharge-and-ending-isolation

European Comission. (2020). EU health preparedness: Recommendations for a common EU approach regarding isolation for COVID-19 patients and quarantine for contacts and travellers. Retrieved from https://ec.europa.eu/health/sites/default/files/preparedness_response/docs/hsc_quarantine-isolation_recomm_en.pdf

Gonçalves, A., Bertrand, J., Ke, R., Comets, E., de Lamballerie, X., Malvy, D.,... Guedj, J. (2020). Timing of Antiviral Treatment Initiation is Critical to Reduce SARS-CoV-2 Viral Load. *CPT Pharmacometrics Syst Pharmacol, 9*(9), 509-514. doi:10.1002/psp4.12543

He, X., Lau, E. H. Y., Wu, P., Deng, X., Wang, J., Hao, X.,... Leung, G. M. (2020). Temporal dynamics in viral shedding and transmissibility of COVID-19. *Nature Medicine, 26*(5), 672-675. doi:10.1038/s41591-020-0869-5

Larremore, D. B., Wilder, B., Lester, E., Shehata, S., Burke, J. M., Hay, J. A.,... Parker, R. (2021). Test sensitivity is secondary to frequency and turnaround time for COVID-19 screening. *Science Advances, 7*(1), eabd5393. doi:10.1126/sciadv.abd5393

Miyakawa, K., Funabashi, R., Yamaoka, Y., Jeremiah, S. S., Katada, J., Wada, A.,... Ryo, A. (2021). SARS-CoV-2 antigen rapid diagnostic test enhanced with silver amplification technology. *medRxiv*, 2021.2001.2027.21250659. doi:10.1101/2021.01.27.21250659

Quilty, B. J., Clifford, S., Hellewell, J., Russell, T. W., Kucharski, A. J., Flasche, S.,... Davies, N. G. (2021). Quarantine and testing strategies in contact tracing for SARS-CoV-2: a modelling study. *The Lancet Public Health*. doi:10.1016/S2468-2667(20)30308-X

Tu, Y.-P., Jennings, R., Hart, B., Cangelosi, G. A., Wood, R. C., Wehber, K.,... Berke, E. M. (2020). Swabs Collected by Patients or Health Care Workers for SARS-CoV-2 Testing. *New England Journal of Medicine*. doi:10.1056/NEJMc2016321

van Kampen, J. J. A., van de Vijver, D. A. M. C., Fraaij, P. L. A., Haagmans, B. L., Lamers, M. M., Okba, N.,... van der Eijk, A. A. (2021). Duration and key determinants of infectious virus shedding in hospitalized patients with coronavirus disease-2019 (COVID-19). *Nature Communications, 12*(1), 267. doi:10.1038/s41467-020-20568-4

Wölfel, R., Corman, V. M., Guggemos, W., Seilmaier, M., Zange, S., Müller, M. A.,... Wendtner, C. (2020). Virological assessment of hospitalized patients with COVID-2019. *Nature, 581*(7809), 465-469. doi:10.1038/s41586-020-2196-x

World Health Organization. (2020, 2020). Criteria for releasing COVID-19 patients from isolation: scientific brief. Retrieved from https://apps.who.int/iris/handle/10665/332451

Wyllie, A. L., Fournier, J., Casanovas-Massana, A., Campbell, M., Tokuyama, M., Vijayakumar, P.,... Ko, A. I. (2020). Saliva or Nasopharyngeal Swab Specimens for Detection of SARS-CoV-2. *New England Journal of Medicine, 383*(13), 1283-1286. doi:10.1056/NEJMc2016359

Yang, Q., Meyerson, N. R., Clark, S. K., Paige, C. L., Fattor, W. T., Gilchrist, A. R.,... Sawyer, S. L. (2021). Saliva TwoStep for rapid detection of asymptomatic SARS-CoV-2 carriers. *medRxiv*, 2020.2007.2016.20150250. doi:10.1101/2020.07.16.20150250